# Unleashing the power of novel conditional generative approaches for new materials discovery

## Abstract

For a very long time, computational approaches to the design of new materials have relied on an iterative process of finding a candidate material and modeling its properties. AI has played a crucial role in this regard, helping to accelerate the discovery and optimization of crystal properties and structures through advanced computational methodologies and data-driven approaches. To address the problem of new materials design and fasten the process of new materials search, we have applied latest generative approaches to the problem of crystal structure design, trying to solve the inverse problem: by given properties generate a structure that satisfies them without utilizing supercomputer powers. In our work we propose two approaches: 1) conditional structure modification: optimization of the stability of an arbitrary atomic configuration, using the energy difference between the most energetically favorable structure and all its less stable polymorphs and 2) conditional structure generation. We used a representation for materials that includes the following information: lattice, atom coordinates, atom types, chemical features, space group and formation energy of the structure. The loss function was optimized to take into account the periodic boundary conditions of crystal structures. We have applied Diffusion models approach, Flow matching, usual Autoencoder (AE) and compared the results of the models and approaches. As a metric for the study, physical pymatgen matcher was employed: we compare target structure with generated one using default tolerances. So far, our modifier and generator produce structures with needed properties with accuracy 41% and 82% respectively. To prove the offered methodology efficiency, inference have been carried out, resulting in several potentially new structures with formation energy below the AFLOW-derived convex hulls.

## 1 Introduction

The search for novel materials with specified properties has been a cornerstone of scientific exploration for decades. From the discovery of semiconductors revolutionizing electronics to the development of superalloys enhancing aerospace technologies, the synthesis of new materials has continually propelled technological advancements.

However, traditional methods for material discovery often employ exhaustive trial and error experimental approaches. In turn, computational efforts, relying on density functional theory (DFT)[1] approaches, usually require huge amounts of computing power. In this regard, automatic descriptor generators[2], GNNs[3][4] and transferable GNN models [5] fueled combination of these methods and machine learning (ML) approaches. In particular, the

Submitted to 38th Conference on Neural Information Processing Systems (NeurIPS 2024). Do not distribute.

utilization of generative machine learning models, such as Variational Autoencoder[6] and GANs[7], presents a paradigm shift in how crystal structures are generated and optimized. By harnessing the power of data-driven approaches, we can navigate the vast landscape of possible crystal structures with unprecedented efficiency and precision.

Recent advancements in the field of materials discovery have yielded promising results through various innovative approaches. For instance, FTCP[6] utilizes Autoencoders for uncovering new materials, while CubicGAN[7] leverages GANs for the discovery of cubic crystal materials. Additionally, Physics Guided Crystal Generative Model (PGCGM)[8] has introduced a method for generating crystal structures based on specific space groups encoding. DP-CDVAE[9] is a model, that combines VAE and diffusion approaches. MatterGen[10] employed equivariant GNNs as score matching function in diffusion processes for crystal structure generation.

One of the most discussed frameworks is GNoME[11] that has made most recent and large advancements in the field of the new materials discovery employs a sophisticated pipeline to discover new materials, particularly focusing on inorganic crystals. This allows for the discovery of innovative materials beyond known structures.

After generating candidate structures through both pipelines, GNoME evaluates their stability by predicting their formation energies. Based on the comparison of the obtained formation energy with those of the known competing phases (i.e. stability assessment), the model selects the most promising candidates for further evaluation using known theoretical frameworks.

The question of the completeness of chemical space arises due to two main concerns with GNoME-derived stable structures. Firstly, they mostly contain three or more unique elements, while ternary and quaternary structures are less explored than binary compounds. Secondly, the comparison of GNoME-discovered structures to the Materials Project, which has 154,718 materials, is flawed since larger databases like AFLOW, NOMAD, and the Open Quantum Materials Database contain millions of entries. This raises questions about the novelty of the discovered materials.

In this study, we present an end-to-end framework for the generation of crystal structures with specified properties using advanced generative AI techniques. The basis architecture of the models is Autoencoder, enabling encoding and decoding structural representations. Then, the most commonly used generative approaches in image generation were utilized to model probability distribution transformations, and to capture complex underlying structure-property relationships within our dataset: Flow Matching[12], Denoising Diffusion Probabilistic Models(DDPM)[13], and Denoising Diffusion Implicit Models(DDIM)[14]. Through the integration of these techniques, we aim to transcend conventional limitations in materials discovery, paving the way for accelerated predictions of materials with desired properties.

To employ model architectures often used for image/video generation, a matrix representation of crystal structures was developed, containing crucial information such as chemical composition, atomic coordinates, symmetries (space group), and formation energies. Within the approach proposed, it has become important to develop a novel metric for assessing the similarity between generated structures and target configurations. This metric obviates the need for computationally expensive DFT calculations, allowing for rapid validation and refinement of generated structures. Furthermore, we introduce a loss function that accounts for the periodic boundary conditions inherent in crystal lattices, ensuring the fidelity of the generated structures.

Our study explores two distinct approaches for crystal structure prediction: 1) conditional structure modification and 2) conditional structure generation. The former involves optimizing the stability of existing structures by generating more stable polymorphs, while the latter entails the generation of entirely new structures based on user-defined criteria. Through rigorous analysis, we demonstrate the efficacy of our approach in discovering novel materials with desired properties.

Importantly, to validate the utility of our framework, we conducted a series of generation experiments using the Vienna Ab initio simulation package(VASP)[15] as a tool for inference validation. Remarkably, our method facilitated the discovery of 6 structures below the corresponding convex hull. This significant outcome underscores the remarkable potential of

92 our framework in uncovering thermodynamically stable materials, thereby offering promising
93 avenues for advanced materials discovery and design.

## 2 Data, Dataset

### 2.1 Data overview

96 In this study, the AFLOW database[16] was utilized as a source of data on the structures
97 and properties of materials. AFLOW is an extensive and comprehensive database that
98 consolidates a vast array of materials-related information, offering an expansive repository
99 for crystallographic data, computed properties, and various other materials-science-related
100 datasets. AFLOW database contains more than 3.5 million structures.

101 From the extensive collection housed within AFLOW, the focus was narrowed to select only
102 polymorphs, because models are trained to distinguish composition-property and structure-
103 property relations with numerous structures of the same chemical composition. Specifically,
104 the selection process targeted polymorphic structures with 4 to 60 atoms within their unit
105 cells. This criterion aimed to encompass a diverse yet manageable subset of structures,
106 balancing complexity with computational feasibility. By filtering polymorphs based on their
107 atom count, the dataset was balanced.

108 Moreover, in order to decrease the complexity of the data, we have removed all structures
109 containing elements and space groups found in less than 1% of all structures. The entire
110 dataset consisted of more than 85000 polymorph groups including more than 2.1 million
111 structures. The minimum size of group of polymorphs was 7 samples and the maximum one
112 was 71 samples. The total number of space groups was 19 and the total number of chemical
113 species over the dataset was 55. Each structure $S$ in the dataset is described by the following
114 features:

115 • Fractional coordinates of atoms in the lattice basis $X_{coord}$ (has 60 rows with 3
116 coordinates $x, y, z$ each) and $X_{lattice}$ (matrix 3 by 3 constructed of 3 base vectors).
117 Overall matrix X of structure is constructed as

$$\underset{64 \times 3}{X} = concatatenation(\underset{60 \times 3}{X_{coord}}, \underset{1 \times 3}{padding}, \underset{3 \times 3}{X_{lattice}}) \tag{1}$$

• Chemical elements which are presented as a one-hot matrix $elements_{ij}$ of size
$64 \times 103$ (including padding), where ones are positioned at the indices corresponding
to the position of a certain chemical element in the periodic table.

$$elements_{ij} = \begin{cases} 1 & \text{if } i\text{-th atom's element number from the periodic table} = j \\ 0 & \text{otherwise} \end{cases}$$

118 • Elemental property matrix $elementalProperties$ containing 22 chemical features
119 encoding chemical elements obtained from [8]. The properties of each element were
120 calculated using Mendeleev package[17].

121 • Space group $spg$ of a structure. We use the space group encoding method presented in
122 [8], when each space group is represented by a $192 \times 4 \times 4$ matrix, which corresponds
123 to 192 possible symmetry operations.

124 • Structure formation energy $E$

125 • Nsites - number of atoms in a crystal lattice.

### 2.2 Data representation. Modification task

127 The crystal pair sampling strategy involves handling a potential data leakage: possible
128 inclusion of structures from the same polymorph group but with different energies into training
129 and validation subsets. To mitigate this issue, the polymorph group formulas were initially
130 divided into distinct training and validation sets, ensuring a relatively balanced distribution
131 of chemical elements across these subsets. Subsequently, the pairs were categorized into two
132 groups: those with low-energy (lowest energy in polymorph group) targets designated as

$lowestEnergyPairs = (S_i, S_0) \forall i \in [1, ..., structuresNum]$ and those with non-low-energy targets, all structures except the most optimal one, formed as $nonLowestEnergyPairs = (S_i, S_j) | \ i > j > 0$. The validation set was constructed as a subset of $lowestEnergyPairs$. The training set was dynamically constructed every epoch from $lowestEnergyPairs$ and $nonlowestEnergyPairs$, preserving equal numbers of pairs sampled and maintaining a limited count per polymorph group. This strategy ensured a robust separation between training and validation sets, thus preventing data leakage and improving model performance.

Each pair sample $\{S_{init}, S_{target}\} \in pairDataset$ consisted of the information about each structure (hereinafter, we will call them initial and target structures). The following data was used:

- Coordinates and lattice information of initial and target structures $X_{init}, X_{target}$
- Difference in formation energies between initial and target structures $E_{diff} = E_{target} - E_{init}$
- Space group of target structure $spg_{target}$
- Elements matrix $elemetsMatrix$, elemental property matrix $elementalProperties$ and number of sites $numSites$, which are the same for initial and target structure because of identical chemical composition.

The modification task involved transforming the input structure $X_{init}$ into the target structure $X_{target}$.

## 2.3 Data representation. Generation task

In its tern the generation task receives normal or uniform (depends on a model) noise as input from which the structure is generated, which is akin to the image generation processes in computer vision tasks.

For the generation task, an additional dataset was constructed. Data for the generation task is slightly simpler, while it considers only $\{S_{target}\}$. Therefore, the models can be trained on all structures available, rather than pairs. The following data is used:

- Coordinates and lattice information of target structure $X_{target}$
- Formation energy of target structure $E_{target}$
- Space group of target structure $spg_{target}$
- Elements matrix $elemetsMatrix$, elemental property matrix $elementalProperties$ and number of sites $numSites$ of target structure.

# 3 Loss and metrics

## 3.1 Atomic coordinates

The atomic coordinates are represented as a $60 \times 3$ matrix, where each row corresponds to the coordinates of an atom. The L1 loss was utilized during the training of a model for predicting atomic coordinates.

$L_1(preds, target)_i = ||preds_i - target_i||_1 = \sum_{j=1}^{3} |preds_{ij} - target_{ij}|$, where $target$ and $pred$ are target and predicted atomic coordinate matrices.

## 3.2 Lattice

The lattice itself is represented as a 3x3 matrix, where each row signifies a directing basis vector. In this case, we have also used the L1 norm as a loss function.

## 3.3 Periodic boundary condition loss

This section presents an enhanced loss function, designed for the regression model (see Section 5.1), that addresses this challenge by integrating periodic boundary conditions into

the loss calculation, outperforming the conventional L1 loss function. In the field of ML applied to atomic structures, even slight displacement of atomic coordinates is crucial and employing appropriate loss functions that consider the periodic nature of atomic structures increases the flexibility of model predictions.

In the dataset representing atomic structures, it is crucial to acknowledge the presence of atoms residing at various positions within the lattice framework. Certain atoms are positioned at the vertices, edges, or faces of the lattice. According to periodic boundary conditions (PBC), identical atoms in the vicinity of vertices, edges, or faces but also exist in analogous positions across the lattice. Implementation of such an invariance within the loss function helps in effectively capturing periodic pattern of crystals, enhancing the model's capability to learn and predict atomic structures more comprehensively.

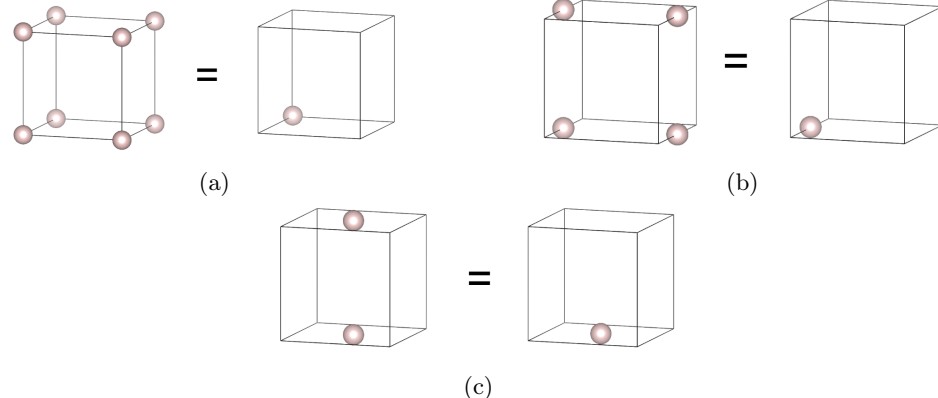

(a)      (b)

(c)

Figure 1: Illustration of atoms at a)vertices, b)edges, and c)faces of lattice under periodic boundary conditions

The loss function is being calculated as minimum of distances from predicted point to the target one taking into account 26 its periodic images (according to PBC) A.4.

The empirical validation of this enhanced loss function showcases its superiority(Figure4) in capturing discrepancies within atomic structures, thus indicating its potential as a robust tool for improving the accuracy of ML models in materials science applications.

## 3.4 Metric

As a metric, we have chosen an analogue of accuracy: the generated structures are compared to the target structures using a specialized matcher, yielding the proportion of structures that successfully pass the matching process. For metric calculation, we employed the Pymatgen StructureMatcher with the default set of parameters ($ltol = 0.2, stol = 0.3, angle\_tol = 5$). Although this approach is less accurate than structure relaxation using ab initio calculations and comparing the structure formation energy with the energy above the hull, it enables model validation to be performed orders of magnitude faster than the traditional method.

## 4 Model

For experiments, a 1d UNet model (see Figure2 (b)) architecture similar to the 2d UNet model described in [18] was utilized along with 2D and 1D convolutional neural networks (CNNs) for the space group and element matrix embeddings, respectively. Based on this model, 3 different training processes have been developed: ordinary regression model, Conditional Flow Matching (CFM)[19] model, and diffusion model.

The model was conditioned (see Figure2 (a)) on the following data: time condition ($t$), the same as in [18], element condition ($el$), formation energy difference condition ($E_{diff}$), and desirable space group ($spg$). $el_{emb}$, $spg_{emb}$ and $E_{diff}$ are concatenated into one embedding

$C_{emb}$. $t$ is fed into the Transformer Positional Encoding Layer and transformed into an embedding $T_{emb}$. The two embeddings: $C_{emb}$ and $T_{emb}$ are then applied into one condition.

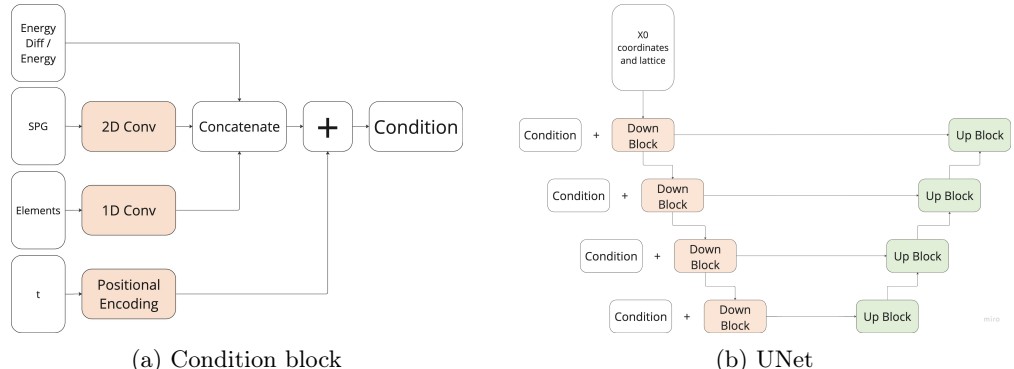

(a) Condition block        (b) UNet

Figure 2: a)Formation of conditions using formation energy, space group, and elemental representation, and b)Schematic depiction of the model architecture

## 5 Methodology

In this work, two approaches are proposed: crystal structure generation and crystal structure modification. For the generation approach, crystal structures are generated from normal or uniform noise and conditioned to $t$, $el$, $E$, $spg$. Within the generation, we employed three algorithms: DDPM, DDIM, and CFM models. For the modification approach, crystal structures are generated by modifying other structures, while conditioning to $el$, $E_{diff}$, $spg$ (and optionally $t$, not used in ordinary regression UNet). For the modification task, we have employed three algorithms: UNet Regression model, diffusion model, based on Palette[20] approach, and CFM model. For the generation task, we have employed four algorithms: diffusion models with DDPM and DDIM samplers, and CFM models on Uniform and Normal noise.

### 5.1 Regression model

During the training stage, the structure coordinates and lattice $x_0$, elements features $el$, space group $spg$ and $E_{diff}$ are used as conditions. The model is trained to return $x_1$ structure coordinates and lattice (Algorithm 1). As for the inference process, one can see the details in the Algorithm 2

### 5.2 Conditional Flow Matching models

CFM is a fast method for training Continuous Normalizing Flows (CNF)[21] models without the need for simulations. It offers a training objective that enables conditional generative modeling and accelerates both training and inference.

The basic way of training CFM model (Algorithm 3) organized as follows: during the training stage, $x_0$ and $x_1$ are sampled from the source distribution and the target distribution respectively, then a linear interpolation $x_t$ is calculated as $x_t = tx_1 + (1-t)x_0$ (exponential moving average between distributions $x_0$ and $x_1$; $t$ is sampled from a uniform distribution $\mathcal{U}(0,1)$), and afterwards pass the $x_t$ and $t$ as inputs to our model $f_\theta$, forcing the model to predict a velocity from the distribution $x_0$ to $x_1$. Therefore, the loss for CFM model is the following: $L_{CFM} = E_{t,x_1,x_0}[||f_\theta(x_t,t) - (x_1 - x_0)||^2] = E_{t,x_1,x_0}[||f_\theta(tx_1 + (1-t)x_0, t) - (x_1 - x_0)||^2]$

For the modification approach, $x_0$ and $x_1$ are both sampled from our dataset distribution according to the sampling strategy for modification mentioned in 2.2. Also, the model is conditioned to $el, spg_1, E_{diff}$, besides $t$ (see Algorithm 4)

243 For the generation approach, we tested two noise distributions for the $x_0$: normal distribu-
244 tion $\mathcal{N}(0,1)$ and uniform noise distribution $\mathcal{U}(0,1)$, which resulted in significantly better
245 performance. The intuition for using uniform distribution instead of normal one was inspired
246 by the diagram of x, y, z coordinate distribution (Figure 3). The model is also conditioned
247 to $el, spg_1, E$, and $t$ (see Algorithm 5)

248 During the sampling stage, we generate $X_1$ structure by the given $X_0$ by solving the following
249 ordinary differential equation (ODE): $dx_t = f_\theta(x_t, t, el, spg_1, E)dt$, beginning with $x_0$. In
250 order to solve the ODE, the Euler method was employed: $x_{t+h} = x_t + hf_\theta(x_t, t, el, spg_1, E)$
251 (Algorithm 6)

## 5.3 Diffusion models

253 In our work, we observe diffusion models. Diffusion models generate samples from a target
254 distribution $x_1$, starting from a source distribution $x_0 \sim \mathcal{N}(0, I)$.

255 During training, these models are trained to reverse a Markovian forward process, which
256 adds noise $x_0$ to the data step by step. Meaning, diffusion models are trained to predict the
257 noise added to the data samples $x_1$. In order to train a model in this setup, the following loss
258 function is used, $L_{simple} = E_{t,x_1,x_0}[||x_0 - f_\theta(\sqrt{\bar{\alpha}_t}x_1 + \sqrt{1 - \bar{\alpha}_t}x_0, t)||^2]$ where $\bar{\alpha}_t = \prod_{s=1}^{t} \alpha_s$
259 and $\alpha_t = 1 - \beta_t$ ($\beta_t$ is the variance by which added noise is being scheduled on each step $t$).

260 Our modification approach is based on Palette, which enables sample-to-sample generation
261 (from noise $\epsilon \sim \mathcal{N}(0,1)$) using $x_0$ structure coordinates and lattice, $el$, $spg_1$, $E_{diff}$ and $t$ as
262 conditions for generation of $x_1$ using the DDPM algorithm. Sampling stage is performed by
263 a backward diffusion process with linear scheduler (see Algorithms 7, 8).

264 For the generation approach ((Algorithm 9), $x_0$ is sampled from a normal distribution and
265 $el$, $spg_1$, $E$, $t$ are fed into the model as conditions. During our experiments, we tested
266 2 approaches: DDPM(Algorithm 10) classic approach and DDIM(Algorithm 11) which
267 results in usage of smaller number of sampling steps in order to speed up the generation
268 process. Moreover, DDIM enables the process of generating samples from random noise to
269 be deterministic.

## 6 Experiment Results

271 All the models presented in tables (Table 1 and Table 2) have been trained with the same
272 hyperparameters and architectures. The metric used is described in Section 3.4. We also
273 provide all experiment details in A.3.

Table 1: Validation metrics on generation task

| DDPM | DDIM | CFM $\mathcal{N}(0,1)$ | CFM $\mathcal{U}(0,1)$ |
|---|---|---|---|
| 0.8074 | **0.82** | 0.482 | 0.8097 |

Table 2: Validation metrics on modification task

| Ordinary Model | Diffusion | CFM |
|---|---|---|
| **0.4148** | 0.3653 | 0.2059 |

## 7 Inference

275 In order to demonstrate a potential of the proposed approaches, we have chosen a chemical
276 composition, containing numerous variations and phases of structures composed of [W, B,
277 Ta] with well-explored convex hull. Structures that lie on the convex hull are considered to
278 be thermodynamically stable, and the ones above it are either metastable or unstable.

## 7.1 Inference pipeline

The proposed testing procedure involves generating test conditions for structures, passing them to the trained generative models, pre-optimizing the generated structures to accelerate the following ab initio calculations, and final relaxation and formation energy calculating using VASP. Although in this work two approaches were proposed: Generation and Modification, the following pipeline has only been applied to generation models, due to the fact, that modification approach is based on structure-polymorphs, which leads to the necessity to have at least one structure with needed composition, which is not always so. That fact makes generation models much more flexible in generation structures not only with needed properties, but also with needed composition. Another advantage of the generation models is value of metric that is two times bigger than in modification tasks. The inference algorithm is as follows:

1. Test Condition Formation:
   - The chosen chemical formulas were utilized for feature extraction of $el$. Three chemical compositions have been used: 1) $Ta_1W_1B_6$, 2) $Ta_1W_2B_5$ and 3) $Ta_2W_1B_5$.
   - We have taken $spg$ presented in the dataset as an additional condition, obtaining 19 space groups.
   - Finally, a set of target formation energies $E$ has been formed. We have carried out three experiments: 1) starting from the energy of the convex hull and decreasing with a step of 0.01 eV/atom, 2) starting from the energy of the convex hull and decreasing with a step of 0.1 eV/atom, and 3) starting from the energy 1 eV/atom less than the energy of the convex hull and decreasing with a step of 0.01 eV/atom. In total, 21 energy values were used for every inference run.
   - Final inference conditions were obtained by making all possible combinations of $spg$ and $E$ for a certain composition $el$

2. Model Inference: The conditions from the step 1 have been put to one of the trained models, resulting in the generation of structures. Two models have been employed: Diffusion approach and Flow matching

3. Pre-Optimization: Following the generation of all structures, each structure has been pre-optimized using the PyMatGen structure relaxation method. The method used m3gnet [22] model with default parameters. PyMatGen pre-optimization contributed to overall speedup of further VASP relaxation.

4. Structure relaxation: Pre-optimized structures were relaxed using VASP (the recommended pseudopotentials, plane wave energy cutoff of 500 eV, Ediif and Ediffg convergence criteria of $10^{-5}$ and $-10^{-2}$ were used).

## 7.2 Inference results

To summarize, 6 experiments have been carried out for two different models and for three formation energy conditionings. Every experiment includes 3*380 structures, per 380 structures for every single chemical composition. The results of experiments can be seen in Table 3

As can be seen, 4 structures were obtained with formation energies significantly lower than those obtained from the AFLOW-derived convex hull. Thus, it can be concluded that this observation indicates the potential stability of the generated structures rather than differences in the computational methods used in this work and during AFLOW generation. Another four structures also have energies below the convex hull, but in the vicinity of it. Thus, their potential stability should be interpreted with caution.

# 8 Data availability

The raw crystal dataset is downloaded from
https://aflowlib.org

Table 3: Inference results. Each matrix element corresponds to either the minimal energy above the hull achieved in an experiment or the energy above the hull of structures with energies below the hull.

| | | Ta1W1B6, meV/atom | Ta1W2B5, meV/atom | Ta2W1B5, meV/atom |
|---|---|---|---|---|
| Diffusion | energy step = 0.01 energy gap = 0 | 10,41 | 3,275 | 13,079 |
| | energy step = 0.1 energy gap = 0 | 11,835 | 3,83 | **−0, 042** |
| | energy step = 0.01 energy gap = 1 | 97,676 | **−1, 409** | 5,539 |
| Flow-Matching | energy step = 0.01 energy gap = 0 | 11,981 | **−0, 483** **−0, 466** **−0, 387** | **−5, 426** |
| | energy step = 0.1 energy gap = 0 | 11,286 | 0,037 | **−5, 497** |
| | energy step = 0.01 energy gap = 1 | 9,529 | **−4, 852** | 1,029 |

## 9 Code availability

The source code for training and inferencing our models can be obtained from GitHub at https://github.com/AIRI-Institute/conditional-crystal-generation

## 10 Conclusion

In this article, we have offered two approaches to generate crystal structures: conditional generation and conditional modification. The first approach is significantly more flexible as it does not require structure-polymorphs, enabling the generation of structures without restrictions on chemical composition, which can be crucial in certain scenarios. Another advantage of the first approach is the simplicity of data preprocessing; it only requires the chemical composition, space group, atom coordinates, and formation energies.

Our methodology has experimentally proven its effectiveness, resulting in four confident potentially new crystal structures with the following energies above the hull: {-1.409, -5.497, -5.426, and -4.852} meV/atom, and four uncertain candidates with energies of {-0.483, -0.466, -0.387, and -0.042} meV/atom. We have demonstrated that conditional generation approaches, commonly used in image generation, are also fruitful in the design of new materials.

Although the proposed methodology demonstrates its efficiency in generating potentially new crystal structures, it has certain limitations. Firstly, the data is represented in a matrix form, which does not account for all possible symmetries of the crystal structures. Secondly, the structures in the dataset range from 4 to 60 atoms per unit cell, with most structures containing fewer than 8 atoms per unit cell. However, to perform well on structures with a large number of atoms per unit cell, the models just should be pretrained on a dataset that includes larger structures.

Furthermore, despite the limited number of experiments(6) and structures generated (7182), we succeeded in identifying hypothetically new structures. We hope that our article will help to reveal the potential of generative AI in design of new materials with targeted thermodynamic properties and inspire other researchers to be part of this innovative journey in materials design. We believe that rapid and efficient generation of novel materials can lead to breakthroughs in various fields such as electronics, pharmaceuticals, and energy storage. This can accelerate technological advancements and make cutting-edge technologies more accessible and affordable.

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

 # A  Appendix section 1

 ## A.1  Distribution of atomic coordinates

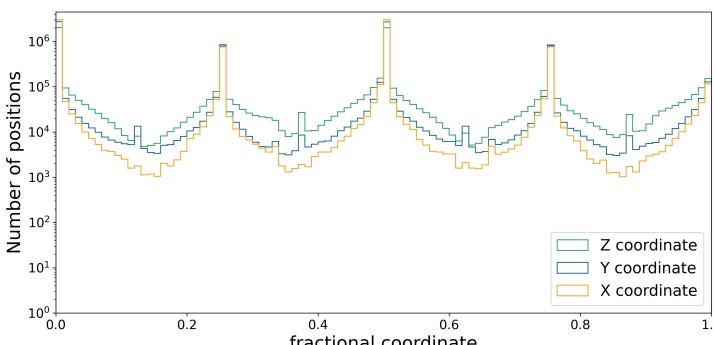

Figure 3: Distribution of the components of fractional atomic coordinates (X, Y, Z)

 ## A.2  Pseudocode

---

**Algorithm 1** Training Regression Modification Model

---

1: repeat
2:      $x_0 \sim q(x_0); x_1 \sim q(x_1); el \sim q(el); spg_1 \sim q(spg_1); E \sim q(E)$
3:      $\mathcal{L} \leftarrow ||x_1 - f_\theta(x_0, t, el, spg_1, E)||$
4:      $\theta \leftarrow Update(\theta, \nabla_\theta \mathcal{L}(\theta))$
5: until converge

---

---

**Algorithm 2** Inferencing Regression Modification Model

---

1: $x_0 \sim q(x_0); el \sim q(el); spg_1 \sim q(spg_1); E \sim q(E)$
2: $x_1 = f_\theta(x_0, t, el, spg_1, E)$
3: return $x_1$

---

---

**Algorithm 3** CFM Training

---

1: repeat
2:      $x_0 \sim q(x_0); x_1 \sim q(x_1)$
3:      $t \sim \mathcal{U}(0, 1)$
4:      $x_t = tx_1 + (1 - t)x_0$
5:      $\mathcal{L}_{CFM} \leftarrow ||f_\theta(x_t, t) - (x_1 - x_0)||$
6:      $\theta \leftarrow Update(\theta, \nabla_\theta \mathcal{L}_{CFM}(\theta))$
7: until converge

---

---

**Algorithm 4** Training CFM for Modification

---

1: repeat
2:      $x_0 \sim q(x_0); x_1 \sim q(x_1); el \sim q(el); spg_1 \sim q(spg_1); E \sim q(E)$
3:      $t \sim \mathcal{U}(0, 1)$
4:      $x_t = tx_1 + (1 - t)x_0$
5:      $\mathcal{L}_{CFM} \leftarrow ||f_\theta(x_t, t, el, spg_1, E) - (x_1 - x_0)||$
6:      $\theta \leftarrow Update(\theta, \nabla_\theta \mathcal{L}_{CFM}(\theta))$
7: until converge

---

---

**Algorithm 5** Training CFM for Generation

---

1: repeat
2:      $x_0 \sim \mathcal{N}(0,1)$ or $x_0 \sim \mathcal{U}(0,1)$
3:      $x_1 \sim q(x_1); el \sim q(el); spg_1 \sim q(spg_1); E \sim q(E)$
4:      $t \sim \mathcal{U}(0,1)$
5:      $x_t = tx_1 + (1-t)x_0$
6:      $\mathcal{L}_{CFM} \leftarrow ||f_\theta(x_t, t, el, spg_1, E) - (x_1 - x_0)||$
7:      $\theta \leftarrow Update(\theta, \nabla_\theta \mathcal{L}_{CFM}(\theta))$
8: until converge

---

**Algorithm 6** Sampling with CFM for Modification or Generation

---

1: $h = \frac{1}{T}$
2: $x_0 \sim q(x_0)$ or $x_0 \sim \mathcal{N}(0,1)$ or $x_0 \sim \mathcal{U}(0,1)$
3: $el \sim q(el); spg_1 \sim q(spg_1); E \sim q(E)$
4: for  do$t = 1, \ldots, T$ do
5:      $x_{t+1} = x_t + hf_\theta(x_t, t, el, spg_1, E)$
6: end for
7: return $x_1$

---

**Algorithm 7** Training DM for Modification

---

1: repeat
2:      $x_0 \sim q(x_0); x_1 \sim q(x_1); el \sim q(el); spg_1 \sim q(spg_1); E \sim q(E)$
3:      $t \sim \mathcal{U}(\{1, \ldots, T\})$
4:      $\epsilon \sim \mathcal{N}(0, I)$
5:      $\mathcal{L}_D \leftarrow ||\epsilon - f_\theta(\sqrt{\bar{\alpha}_t}x_1 + \sqrt{1 - \bar{\alpha}_t}\epsilon, x_0, t, el, spg_1, E)||$
6:      $\theta \leftarrow Update(\theta, \nabla_\theta \mathcal{L}_D(\theta))$
7: until converge

---

**Algorithm 8** Sampling with DM for Modification

---

1: $x_T \sim \mathcal{N}(0, I)$
2: for  do$t = T, \ldots 1$ do
3:      $x_0 \sim q(x_0); x_1 \sim q(x_1); el \sim q(el); spg_1 \sim q(spg_1); E \sim q(E)$
4:      $z \sim \mathcal{N}(0, I)$ if $t > 1$ else $z = 0$
5:      $x_{t-1} = \frac{1}{\sqrt{\alpha_t}}(x_t - \frac{1-\alpha_t}{\sqrt{1-\bar{\alpha}_t}}f_\theta(x_t, x_0, t, el, spg_1, E)) + \sqrt{1 - \alpha_t}z$
6: end for
7: return $x_1$

---

**Algorithm 9** Training DM for Generation

---

1: repeat
2:      $x_1 \sim q(x_1); el \sim q(el); spg_1 \sim q(spg_1); E \sim q(E)$
3:      $t \sim \mathcal{U}(\{1, \ldots, T\})$
4:      $\epsilon \sim \mathcal{N}(0, I)$
5:      $\mathcal{L}_D \leftarrow ||\epsilon - f_\theta(\sqrt{\bar{\alpha}_t}x_1 + \sqrt{1 - \bar{\alpha}_t}\epsilon, t, el, spg_1, E)||$
6:      $\theta \leftarrow Update(\theta, \nabla_\theta \mathcal{L}_D(\theta))$
7: until converge

---

**Algorithm 10** DDPM Sampling

---

1: $x_T \sim \mathcal{N}(0, I)$
2: for  do$t = T, \ldots 1$ do
3:      $x_1 \sim q(x_1); el \sim q(el); spg_1 \sim q(spg_1); E \sim q(E)$
4:      $z \sim \mathcal{N}(0, I)$ if $t > 1$ else $z = 0$
5:      $x_{t-1} = \frac{1}{\sqrt{\alpha_t}}(x_t - \frac{1-\alpha_t}{\sqrt{1-\bar{\alpha}_t}}f_\theta(x_t, t, el, spg_1, E)) + \sqrt{1 - \alpha_t}z$
6: end for
7: return $x_1$

---

---
**Algorithm 11** DDIM Sampling
---
1: $x_T \sim \mathcal{N}(0, I)$
2: for $\text{do} t = T, \ldots 1$ with step C do
3:     $x_0 \sim q(x_0); x_1 \sim q(x_1); el \sim q(el); spg_1 \sim q(spg_1); E \sim q(E)$
4:     $z \sim \mathcal{N}(0, I)$ if $t > 1$ else z = 0
5:     $x_\theta = f_\theta(x_t, t, el, spg_1, E)$
6:     $x_{t-1} = \sqrt{\alpha_{t-1}}(\frac{x_t - \sqrt{1-\alpha_t}x_\theta}{\sqrt{\alpha_t}}) + \sqrt{1 - \alpha_{t-1} - \sigma_t^2}x_\theta + \sigma_t z$
7:
8: end for
9: return $x_1$
---

### A.3 Experiment Details

All the experiments use the same hyperparameters for the model:

- num_res_blocks = 7
- attention_resolution = (1, 2, 4, 8)
- model_channels = 128

In all the experiments models are trained with the same training parameters:

- optimizer = Adam
  - betas = (0.9, 0.999)
  - eps = 1e-08
  - weight_decay = 0
- batch_size = 256
- epochs = 400
- learning_rate = 1e-4
- lr_warmup_steps = 500
- random_state = 42

An important note, that all our experiments have been conducted in mixed precision in fp16.

Generation task: Diffusion Model (DDPM):

- num_train_timesteps = 1000 (diffusion process discretization)
- beta_start = 0.0001
- beta_end = 0.02
- num_inference_steps = 100
- beta_schedule = "squaredcos_cap_v2" (cosine)

Diffusion Model (DDIM):

- num_train_timesteps = 1000 (diffusion process discretization)
- beta_start = 0.0001
- beta_end = 0.02
- num_inference_steps = 100
- beta_schedule = "squaredcos_cap_v2" (cosine)

Flow Matching $x_0 \sim \mathcal{N}(0, 1)$:

- num_inference_steps = 100

Flow Matching $x_0 \sim \mathcal{U}(0, 1)$:

466    • num_inference_steps = 100

467 Modification task:

468 Regression UNet:

469    • num_inference_steps = 1

470 Diffusion Model:

471    • num_train_timesteps = 1000 (diffusion process discretization)
472    • beta_start = 0.0001
473    • beta_end = 0.02
474    • num_inference_steps = 100
475    • beta_schedule = "squaredcos_cap_v2" (cosine)

476 Flow Matching:

477    • num_inference_steps = 100

478 **A.4   PBC Loss details**

479 The PBC loss function operates through several steps:

1. Vertices evaluation: If the target coordinate of the atom is lattice vertex (all 3 coordinates $x, y, z$ are equal to 1 or 0), then loss between prediction point $preds_i$ and target point $target_i$ is being calculated using following formula:

$$L_{vertex}(preds_i, target_i) = \min_{v \in vertices} ||preds_i - v||,$$

480 where $vertices$ is a set of 8 possible positions according to PBC ($\{0, 0, 0\}$, $\{0, 0, 1\}$,
481 ..., $\{1, 1, 1\}$).

482 2. Edges evaluation: If the target coordinate of the atom is located on lattice edge (two
483    coordinates are equal to 1 or 0 and one is not). For example, a lattice edge atom
484    at point $\{0, 1, 0.3\}$ has identical atoms at points $\{0, 0, 0.3\}, \{1, 0, 0.3\}, \{1, 1, 0.3\}$. As
485    we can see, in this example $z$-coordinate is fixed but $x$ and $y$ are exchangeable.
486    Therefore, if the target point is represented as $\{x, y, z\}$, we can use the following
487    formula:

$$L_{edge}(preds_i, target_i) = \min_{e \in edgePoints} ||preds_i - e||,$$

488 where $edgePoints$ is a set of 4 possible positions according to PBC.

489    • Case of fixed point $x$: $edgePoints = \{\{x, 0, 0\}, \{x, 0, 1\}, \{x, 1, 0\}, \{x, 1, 1\}\}$
490    • Case of fixed point $y$: $edgePoints = \{\{0, y, 0\}, \{0, y, 1\}, \{1, y, 0\}, \{1, y, 1\}\}$
491    • Case of fixed point $z$: $edgePoints = \{\{0, 0, z\}, \{0, 1, z\}, \{1, 0, z\}, \{1, 1, z\}\}$

492 3. Sides evaluation: If the target coordinate of the atom is located on lattice side (one
493    coordinate is equal to 1 or 0 and two are not). For example, lattice side atom at
494    point $\{0, 0.5, 0.3\}$ has identical atom at point $\{1, 0.5, 0.3\}$. In this example $y$ and
495    $z$ coordinates are fixed but $x$ is exchangeable. Therefore, if the target point is
496    represented as $\{x, y, z\}$, we can use the following formula:

$$L_{size}(preds_i, target_i) = \min_{s \in sidePoints} ||preds_i - s||,$$

497 where $sidePoints$ is a set of 2 possible positions according to PBC.

498    • Case of exchangeable point $x$: $sidePoints = \{\{0, y, z\}, \{1, y, z\}\}$
499    • Case of exchangeable point $y$: $sidePoints = \{\{x, 0, z\}, \{x, 1, z\}\}$
500    • Case of exchangeable point $z$: $sidePoints = \{\{x, y, 0\}, \{x, y, 1\}\}$

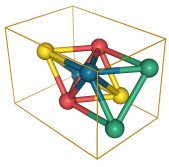
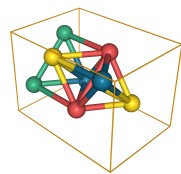

(a) Target structure          (b) Prediction

Figure 4: Example of using PBC-aware loss. The depicted structures (Mo2Nb2Ta2W2) are visually different, but in fact they are exact the same. It is confirmed by insignificant value of PBC-aware loss

4. Points, which don't belong to the groups above, are processed using the default loss function.

Since the $\min(x_1, x_2, ..., x_n)$ function is undifferentiable at multiple points ($x_i = x_j \; \forall i \neq j$), it makes a loss function to have a more complicated surface. Therefore, we used a norm function with order $k \to -\infty$ which is differentiable at all points as a replacement.

$$min_{diff}(x_1, x_2, ..., x_n) = (\sum_{i=1}^{n} |x_i|^k)^{\frac{1}{k}} \quad k \longrightarrow -\infty$$

Therefore, overall PBC-aware loss for a structure is represented as:

$$L_{PBC}(preds, target) = \sum_{i=1}^{n} \mathbb{I}(target_i \text{ is vertex point})L_{vertex}(preds_i, target_i)$$
$$+\mathbb{I}(target_i \text{ is edge point})L_{edge}(preds_i, target_i)$$
$$+\mathbb{I}(target_i \text{ is side point})L_{side}(preds_i, target_i)$$
$$+\mathbb{I}(target_i \text{ is usual point})L_2(preds_i, target_i)$$

As the count of atoms varies across different structures, the $L_{PBC}$ metric tends to yield higher values for structures featuring a larger number of atoms. Thus, it is important to normalize the loss function with the number of atoms in the structure if it would be used in batches with structures with different number of atoms. Therefore, a PBC-aware loss for a batch of structures is formulated as:

$$L_{batchPBC}(batchPreds, batchTargets) =$$
$$\sum_{i=1}^{batchSize} \frac{1}{numSites_i} L_{PBC}(batchPreds_i, batchTargets_i)$$

## Compute resources

For our computational needs in model training and inference, we deployed a total of three GPU servers with the following configurations:

Server 1:

- GPU: NVIDIA A100/80G
- CPU: 8vCPU of Intel(R) Xeon(R) Gold 6248R @ 3.00 GHz
- RAM: 64Gb

Server 2:

- GPU: NVIDIA V100 (32GB)
- CPU: 8vCPU of Intel(R) Xeon(R) Gold 6278C @ 2.60 GHz
- RAM: 64Gb

Server 3:

- GPU: NVIDIA V100 (32GB)
- CPU: 8vCPU of Intel(R) Xeon(R) Gold 6278C @ 2.60 GHz
- RAM: 64Gb

Every model training time consumed up to 2 weeks employing computing power of one GPU server.

For the ab-initio calculations implemented in VASP, we deployed a total of 5 identical CPU servers with the following configurations:

- CPU: 64vCPU of Intel(R) Xeon(R) Gold 6278C CPU @ 2.60GHz
- RAM: 256Gb

Structure relaxation with VASP for all six experiments mentioned in Table 3 took more that 180 thousand CPU hours. Computing power of all CPU servers was employed.

