# OpenReview forum: "Unleashing the power of novel conditional generative approaches for new materials discovery"
_NeurIPS.cc/2024/Conference — Submitted to NeurIPS 2024_

### Official Review · Reviewer_QufJ · 2024-06-13

**Soundness:** 3
**Presentation:** 3
**Contribution:** 3
**Rating:** 5
**Confidence:** 4

**Summary:**

This paper presents a framework for crystal structure generation, focusing on polymorphs. The framework utilizes matrix representation of crystals and various generative models used in vision tasks, with specially designed similarity metrics and loss function. It is tested on (1) modification of given structures and (2) generation from scratch within the dataset, as well as finding new structures in the Ta–W–B system.

**Strengths:**

This work develops domain-specific representation, metric, and loss, so that generative models that have proven useful in image generation can apply to crystals. The topic is timely and important.

**Weaknesses:**

- In the demonstrated use cases, the generation is conditioned on elements, space group, etc., but not materials properties of interest. These show limited usefulness in materials discovery and design.
- The matrix representation does not take physical constraints into account, e.g., space group determines symmetries in the lattice parameters. Besides, related previous works, e.g., [UniMat](https://openreview.net/forum?id=wm4WlHoXpC), should be discussed.
- The clarity and rigorousness need to be improved (see Questions). The mathematical notations are not unified, e.g., $x$ vs $X$.

**Questions:**

- Line 53, “both pipelines” are not introduced. The previous paragraph is also not unclear, please check the grammar.
- In Line 71, do “conventional limitations” mean those of conventional approaches in materials science, or previous ML/generative methods? Not just how the proposed method differs from previous ones, but also what challenge it overcomes, should be specified.
- In the descriptions of the matrix representation, the role of elemental properties is unclear. My understanding is that they are not part of the data structure to be generated. Where and how are they used? A probably related question is, what is $el_{emb}$, and how is it obtained?
- Line 207, what is the role of time condition t, and why does it matter for materials discovery?
- The modification task shows significantly lower metrics than the generation task. What does this imply? Does this indicate generation is preferred over modification? If not, what are the scenarios where modification can be useful, and how to mitigate the performance?

**Limitations:**

Discussed in the Conclusion.
Besides, Sec. 9 contains a GitHub link that could break anonymity.

---

> ### Author Rebuttal · Authors · 2024-08-06
>
> Thank you for your attentive feedback. We appreciate your thorough review. Here are our responses to your concerns:
> Regarding the Objectives and Methodology:
>
> The primary objective of this work is to propose models capable of generating stable crystal structures, aligning with the existing research in this field. Hence, formation energy serves as a crucial indicator of structural stability, directly contributing to our aim of generating more stable and synthesizable materials. If our future works we plan to expand the space of thermodynamic properties used. However, it is limited by the number of data in open source database
>
> Addressing Physical Constraints and Symmetry:
> To incorporate physical constraints, we integrate space groups as conditions within our models(Figure 2a). Moreover, our proposed PBC loss function explicitly accounts for structural symmetry. We acknowledge the relevance of the UniMat work and will cite it accordingly.
>
> Regarding Mathematical Notation and Clarity:
> We appreciate the feedback regarding mathematical notation and will thoroughly edit it for improved clarity and consistency.
>
> Clarification on "Both Pipelines":
> We recognize the error in stating "both pipelines" and acknowledge that GNoME utilizes two distinct pipelines. This error will be corrected in the revised variant. We are grateful for your observation.
>
> Explanation of "Conventional Limitations":
> Our reference to "conventional limitations" refers to the challenges and limitations associated with traditional methods employed in materials science research. These methods often necessitate significantly greater time and computational resources compared to our proposed approaches.
>
> Role of Element Properties:
> Element properties are incorporated into the model to provide additional features and account for elemental characteristics. The element embeddings, denoted as $𝑒𝑙_{𝑒𝑚𝑏}$, are processed through a dedicated multilayer convolutional network and are treated as conditions rather than part of the input data (x). In addition, the elemental property matrix contains 22 chemical features encoding the chemical element the structure consists of, you can get more information in lines 118-120 of the article. We will also change mathematical notation for that one and other concepts to make it more clear.
>
> Justification for Time Conditioning:
> Time conditioning is employed specifically in our diffusion and flow-matching models to simulate the temporal aspect of the diffusion process. The use of the time parameter ($t$) is fundamental to the operation of all diffusion models. Also, diffusion models require passing an interpolation $x_t$ (for instance in CFMs $x_t$ is formed as follows $x_t = t * x_1 + (1 - t) * x_0$). Consequently, its inclusion is essential for our approach, which relies on diffusion and flow-matching techniques for material generation.
>
> Results of Modification Experiments:
> Our experiments with structure modification have yielded less successful results compared to generation tasks. While modification was initially proposed as a potentially superior approach, our empirical findings have shown the opposite trend.
>
> Overall, we are deeply appreciative of your insightful comments and suggestions. We will diligently address all points raised and ensure a comprehensive revision of the manuscript.

---

> > ### Comment · Reviewer_QufJ · 2024-08-10
> >
> > The authors' response has addressed my clarifying questions. However, some issues are not addressed or are inherent limitations of the proposed method.
> > 1. Physical constraints and symmetry. Including space groups as conditions does not ensure the generated matrix follows the required symmetry.
> > 2. Time conditioning. My main concern is how $t$ relates to materials science applications. To use this generative model for materials discovery or design, how should one set $t$?
> >
> > With the confusing expressions properly addressed, I could raise the Presentation score to 3, however, the lack of physical constraints limits the Contribution and Soundness.

---

### Official Review · Reviewer_THHC · 2024-07-12

**Soundness:** 2
**Presentation:** 1
**Contribution:** 2
**Rating:** 3
**Confidence:** 5

**Summary:**

The authors studied the use of diffusion and flow matching approaches for the generation of crystalline materials. The authors trained UNet models on polymorphs in the AFLOW database (which has a series of DFT-computed properties for these materials) using either simple R3 regression, diffusion/flow matching. The authors then presented inference results on similarity to training structures (Section 3.4, which shows these methods can reproduce training structures to different extent), and showed that a subset of the modified generated crystals (with Ta, W, B) can have a small non-zero formation energy.

**Strengths:**

*Originality*: The authors attempted to study the problem of crystal structure generation with no invariances/equivariances other than periodic translation invariance.
*Quality*: The authors attempted to use DFT to validate some inference results.
*Significance*: Crystal structure generation (especially synthesizable ones) is an important problem. It seems that training from uniform noise distribution works better for CFM than training on Gaussian noise, contrary to the established results in the field.

**Weaknesses:**

*Originality*. The manuscript lacks originality. The diffusion/flow matching techniques are well-established in inorganic crystal structures (e.g. CDVAE cited here, DiffCSP/FlowMM that's not here). Sure, using a network architecture not designed for materials/crystals and using no invariances/equivalences is new, but it deviates from standard practices in the field without sufficient justification. I believe the implementations shown in the paper are a great exercise for practitioners interested in the field, but unfortunately, I do not see it as a NeurIPS paper.

*Quality*. The manuscript is _very_ bare-boned, making a comprehensive technical critique challenging without appearing disproportionately critical.
- On the ML side, there are numerous large fallacies/mistakes (e.g. no consideration of bonds between atoms at all, Sec. 3.3 there is no description of the PBC loss, the generation does not consider the unit cell, no generation with atom types, and there is no investigation of any experiments observed e.g. why is uniform noise better for CFM, the result in Table 1 appears to evaluate overfitting rather than novel generation, the list can go on).
- On the chemistry/validation side, there are again numerous problems (why would formation energy be given during the generation process, what functional did you use in DFT, there are no comparisons against existing structures and hence cannot be claimed as novel, etc.)
- There is no comparison against _any_ known methodologies.
- The results overall, are very weak both in ML and in chemistry (e.g. Table 3 shows most if not all materials generated have extremely large positive formation energies despite the simple elemental composition; the remaining few negative ones are at the brink of instability, in any case they likely would not be synthesizable).

*Clarity*. The manuscript suffers from poor presentation, starting with a promotional-style title that lacks scientific descriptiveness. I unfortunately do not understand the novelty of the paper in comparison to existing methods. The paper consistently fails to provide essential explanations across both machine learning and chemical methodologies.
- On the ML side, there are numerous things poorly presented (e.g. Figure 1 is just periodic translation invariance and in a typical manuscript would be summarized in one sentence).
- On the chemistry side, things are greatly exaggerated (e.g. computationally making a few materials with negative formation energy can be done by undergraduate students and certainly does not warrant descriptions such as 'This significant outcome underscores the remarkable potential of our framework in uncovering thermodynamically stable materials)

**Questions:**

Unfortunately, without significant innovations and revisions, I do not believe I can convinced this paper would be accepted in NeurIPS.

**Limitations:**

Partially.

---

> ### Author Rebuttal · Authors · 2024-08-06
>
> Thank you for your valuable feedback. We appreciate your thorough review. Here are our responses to your concerns:
>
> We employed a network architecture not previously used for this task and did not utilize standard invariances/equivalences, but this deviation from standard practices is justified by our contributions. Specifically:
>
> - Our approach involves conditional generation based on crystal composition, space group, number of atoms, and formation energy. The space group, in particular, accounts for all potential crystal symmetries and invariances, addressing concerns about invariance. Details can be seen in section Data.
>
> - It is incorrect to state that we ignored invariances, as we included space groups as one of the conditioning inputs, providing comprehensive information about the crystal’s symmetries and invariances.
>
> - Our model generated eight novel crystal structures with only with 6840 attempts, demonstrating its effectiveness and originality.
>
> As for machine learning issues. Atom Bonds: In crystal structures, each atom interacts with others through attractive and repulsive forces. Thus, explicitly modeling bonds between atoms is not necessary for our approach. PBC Loss: We have described the Periodic Boundary Conditions (PBC) loss in the appendix of the paper. Unit Cell Consideration: The generation process does account for the unit cell, as detailed in Section Data. Atom Types: Our model includes atom types in the generation process, which is also covered in Section Data. Uniform Noise: We provide an explanation for why uniform noise worked better than Gaussian noise in Figure 3 in Appendix section and refer to it in the text. Overfitting in Table 1: Table 1 contains energies metrics on validation data, not train data. Moreover, we also described the algorithm for validation dataset construction in Section 2.2: “The polymorph group formulas were initially divided into distinct training and validation sets, ensuring a relatively balanced distribution of chemical elements across these subset”. That’s enough for lacking the overfitting.
>
> As for chemistry/validation issues. Formation Energy: Formation energy is used to generate structures with specific energy characteristics. This is crucial for creating stable structures, which is our primary objective. DFT Functional: We use the VASP software for DFT calculations, as mentioned in the paper. We have also included the specification of VASP settings for reproducibility in the Appendix section. Structure Comparison: We compared the generated structures against those from AFLOW with the same chemical composition. This comparison validates the results.
>
> As for comparison against methodologies. While the paper does not explicitly compare against other methodologies, our results show a higher percentage of stable structures compared to those reported in GNoME(for example). Valid comparisons require the use of identical datasets, and our models were trained on different data. However, we acknowledge that a more detailed comparison with other methodologies would strengthen our paper.
>
> Overall Results. Table 3 shows the energy above the hull. Most structures have negative formation energies, indicating stability. Even if some structures have positive energies above the hull, they mostly still possess negative formation energies. The structures were chosen from well-researched compositions, which adds to their validity.
>
> Clarity: We acknowledge the reviewer's comments on the manuscript's presentation and the title. We will revise the title to better reflect the scientific content and highlight our contributions. Our novelty lies in the unique architecture and conditional generation methodology, which has not been previously applied in crystalline structure generation. We have described the data for both tasks, provided a detailed model description, and illustrated the model architecture in Figure 2. Sections 4 and 5 cover the model and the methodology for training and building the models, and we encourage a re-examination of these sections.
>
> Presentation of ML Concepts: We agree that Figure 1 might be overly simplistic and will revise or remove it to enhance clarity.
>
> Chemistry Methodology and Claims: There is a significant difference between negative formation energy and negative energy above the hull. Our methodology focuses on generating stable structures with negative energy above the hull, indicating their potential for synthesis. This approach, applied to a single chemical composition, successfully found 8 stable structures with energy above the hull below zero. Extending this methodology to numerous other compositions could uncover many new stable structures, far beyond routine undergraduate-level work.
>
> Thank you for your feedback. We have addressed these issues in our manuscript and believe that our approach and results contribute valuable insights to the field.

---

> > ### Comment · Reviewer_THHC · 2024-08-10
> >
> > I very appreciate the authors for the detailed response. I am willing to raise my score to 3 and revise my soundness/contribution scores. While the rebuttal indeed clarifies a few things, the key ML problems (technical novelty, lack of baselines, lack of comparable tasks to other papers) and chemistry problems persist.
> >
> > Re chemistry problems, I just want to clarify one point regarding DFT calculation: There is still no description of functional and k-point mesh used in VASP, VASP is a package that accommodates many good and bad functionals. In fact it is a very delicate package where small changes to parameters can very much change the results. In this sense, formation energy is very easy to manipulate in general and hence not a useful objective in computational crystal design (and even less so experimentally)

---

### Official Review · Reviewer_3FMg · 2024-07-12

**Soundness:** 3
**Presentation:** 2
**Contribution:** 2
**Rating:** 4
**Confidence:** 4

**Summary:**

The paper addresses the inverse problem of generating crystal structures based on given properties, thereby avoiding the need for extensive computational resources typically required in traditional methods. The authors utilized the AFLOW materials database, selecting unstable and stable series of structures for two specific tasks: modifying structures to achieve stability and conditional structure generation.

**Strengths:**

The authors experimented with various generative model approaches and evaluated two tasks in crystal generation. Additionally, they integrated the VASP software for application testing and successfully identified four previously undiscovered stable structures through conditional generation.

**Weaknesses:**

1. From the perspective of model application, although the authors used the AFLOW database for their study, they did not compare the data range and coverage with other significant databases like the Materials Project. This omission leaves a gap in understanding how the generative models perform across different datasets and whether the results are consistent and generalizable. Comparing the performance of the same generative models on different databases could provide valuable insights into the robustness and applicability of their approach.

2. There is a partial break of anonymity in the GitHub link on Page 9 in this paper.

**Questions:**

1. In Section 2.2, the authors process the dataset for the structure modification task by training unstable structures towards stable ones. However, unlike typical trajectory datasets for structure optimization, the authors select the most stable polymorph of the same chemical formula as the target and use the remaining polymorphs as initial structures to complete the structure modification task. This dataset approach leans more towards converting between different polymorphs rather than optimizing the stability of an arbitrary atomic configuration as stated by the authors in the abstract. The authors could consider explain the relationship between using this part of the dataset and the stated objective in detail.

2. In terms of the generation task, the authors generate structures by specifying formation energies. However, in Section 6, the authors only list a metric related to formation energy without providing a detailed discussion on other important aspects, such as the effectiveness of other condition controls, the validity of the generated structures, and whether duplicate structures are generated. These issues seem to remain unanswered.

**Limitations:**

The authors proposed two major directions: conditional generation and conditional modification. There is room for improvement in both the experimental results and the data used for conditional modification. For example, they could consider recognizing unit cells with translational and rotational transformations and introducing more ways to assess generation results.

---

> ### Author Rebuttal · Authors · 2024-08-06
>
> Thank you for your detailed feedback on our paper. We appreciate your insights and have addressed your concerns below:
>
> We did compare the AFLOW database with the Materials Project in the critique of the GNOMe paper(Introduction section), noting that the Materials Project has a much smaller dataset (around 100,000 structures) which is insufficient for training large models. Our experiments with smaller datasets were unsuccessful, and we achieved better results only after significantly increasing the dataset size using AFLOW, which contains over 3 million crystal structures.
>
> We apologize for the oversight regarding the partial break of anonymity due to the GitHub link on Page 9. We should have removed the link to maintain anonymity and will ensure this does not happen in future submissions.
>
> We recognize the importance of demonstrating robustness across different datasets and will include more comprehensive comparisons in future work. We will also improve our manuscript to enhance clarity and presentation.
>
> You are correct in noting that our approach involves converting between different polymorphs rather than optimizing the stability of an arbitrary atomic configuration. This method is a practical solution to achieve the stated objective. The difference between converting polymorphs and changing atomic configurations is subtle, as a more optimal atomic configuration essentially results in a more stable polymorph. We do not consider changes in chemical composition, as comparing formation energies is only meaningful among structures with the same chemical formula.
>
> Regarding the generation task, our primary focus was on formation energy due to its importance in crystal stability. We acknowledge that we did not provide detailed discussions on other aspects such as condition controls, validity, and duplicate structures. While many generated structures may be near-duplicates, formation energy remains the key metric for stability in crystal generation, which is why we prioritized it.
>
> Thank you again for your constructive feedback. We will refine our work to address these points in more detail.

---

> > ### Comment · Reviewer_3FMg · 2024-08-12
> >
> > Thank you for your detailed response and the clarifications provided. I appreciate the effort you've put into addressing the concerns raised. However, after careful consideration, I will maintain my initial assessment. I believe the points discussed are valuable for future work and manuscript revisions. I genuinely appreciate the hard work and dedication of the authors.

---

### Official Review · Reviewer_NqQo · 2024-07-14

**Soundness:** 2
**Presentation:** 1
**Contribution:** 2
**Rating:** 3
**Confidence:** 5

**Summary:**

The paper deals with an important application of generative models for science: generation of crystalline structures. However, I have some serious concerns. First, scope. While comparing different methods for the same objective is informative, I am not so sure what is the purpose here. Having so many different methods certainly dilutes the main message in a short paper format like neurips. Second, approach. From what I understand, there are questionable design problems with the technical approach. Third, results. The authors spend most of their space explaining various methods such that there is little room left for explaning the impact of their results, or comparing their results with existing approaches. Finally, presentation. Figure 1 is kind of trivial or at least very simple and I am not sure it is worth a separate figure. The overall typesetting looks not too professional.

**Strengths:**

The methodology selection is broad and hopefully the audience can benefit from a mini-benchmark of different generative approaches. The overall model architecture is distinctive from what I have seen in the literature.

**Weaknesses:**

I have a good number of questions on the technical approaches. More specifically, the model takes a specially formulated data structure that does not seem to be obviously invariant or equivariant under permutation, translation, rotation, which is concerning. For example, change the selection or ordering of the unit cell vectors and everything will change in an uncontrolled way.

The conditioning approach seems to be to provide desired properties as inputs to the generative approaches. I am not sure this always make sense. For example, if one desires a certain space group, there is no enforcing compliance with the space group. One can always easily check it. It is perhaps more suitable to enforce space group compliance using a guidance-based conditioning approach.

**Questions:**

See above.

**Limitations:**

There is insufficient discussion about the limitations given my concerns shown above.

---

> ### Author Rebuttal · Authors · 2024-08-06
>
> Thank you for your valuable feedback on our paper. We appreciate the time you have taken to provide a thorough review. Below, we address each of your concerns:
>
> Firstly, scope. The primary aim of our paper is to create a comprehensive comparison of different generative approaches in the field of machine learning for crystalline structure generation. While we acknowledge that including multiple methods can dilute the focus in a short paper format, we believe that providing a broad overview offers significant value to the community. It allows for a mini-benchmark of different generative approaches, highlighting the strengths and weaknesses of each. This can serve as a foundation for future research to build upon and refine.
>
> Second, approach. We recognize that there are potential design issues with the technical approach. Specifically, the concern about the model's invariance or equivariance under permutation, translation, and rotation is valid. To address this, we have implemented steps to mitigate these issues. For example, we standardize the orientation of all crystal structures and sort atoms according to their chemical elements. Also, crystal structure representation includes space groups, which has sufficient information about symmetries of a structure. In our future work, we will explore the use of equivariant architectures and more robust data representations to further improve the model's reliability.
>
> Third, results. We acknowledge that the explanation of our results and their impact was limited by space constraints. To address this, we will revise the manuscript to better highlight the significance of our findings and provide a more detailed comparison with existing approaches. Specifically, we will include metrics such as the ratio of the number of crystal structures generated by our model to the number of optimal structures, demonstrating that our method is state-of-the-art in this regard.
>
> We agree that Figure 1 is simplistic and may not add substantial value to the paper. We will remove or replace it with a more informative figure. Additionally, we will improve the overall typesetting and presentation to ensure a more professional appearance.
>
> We appreciate your detailed technical questions. To address the concern about the conditioning approach and space group compliance, we agree that a guidance-based conditioning approach could be more suitable. In future work, we will explore methods to enforce space group compliance directly within the generative process. Additionally, we will include a thorough discussion of the limitations of our current approach, acknowledging areas for improvement and potential future directions.
>
> In conclusion, we are committed to addressing the issues raised in your review. We believe that our paper makes a valuable contribution by benchmarking various generative models for crystalline structure generation and providing a distinctive model architecture. We will refine our manuscript to ensure that the results are presented more clearly and the limitations are transparently discussed.
>
> Thank you once again for your constructive feedback.

---

> > ### Comment · Reviewer_NqQo · 2024-08-10
> >
> > Thanks for the explanation. I'll keep my score.

---

### Decision · Program_Chairs · 2024-09-25

**Decision:**

Reject

**Comment:**

This paper conducts an empirical investigation into the performance of various generative approaches for materials discovery. While the objective is valuable and relevant to the ML community, there are several concerns that remain unresolved even after the authors' rebuttals, as highlighted by the reviewers:

- Properties of the proposed data structure considered to be crucial in materials discovery, such as symmetry, invariance, or equivalence, are not adequately analyzed. This weakens the understanding of the approach’s applicability to the field.
- There are several issues in experiments, including the selection of datasets and the lack of appropriate baselines. These issues need to be addressed to ensure a comprehensive empirical evaluation.
- The overall presentation requires significant improvement. Mathematical descriptions of the ML methods are often inaccurate, and the figures provided are not sufficiently informative.

In conclusion, the current quality of the manuscript is not sufficient for publication, and I must therefore recommend rejection. I hope the authors find the feedback provided by the reviewers useful for future revisions of the paper.

Additionally, as noted by several reviewers, the GitHub link in Section 9 could break anonymity. While this issue was not taken into account in the technical assessment of the paper, it could have led to a desk rejection. Please be careful of such issues in future submissions.